# Innovative Injection Molding Process for the Fabrication of Woven Fabric Reinforced Thermoplastic Composites

**DOI:** 10.3390/polym14081577

**Published:** 2022-04-13

**Authors:** Euichul Jeong, Yongdae Kim, Seokkwan Hong, Kyunghwan Yoon, Sunghee Lee

**Affiliations:** 1Department of Mechanical Engineering, Dankook University, Yongin 16890, Korea; euicha0817@kitech.re.kr; 2Department of Molding & Metal Forming R&D, Korea Institute of Industrial Technology, Bucheon 14441, Korea; ydkim@kitech.re.kr (Y.K.); skhong@kitech.re.kr (S.H.)

**Keywords:** injection molding, reinforced thermoplastic, composite, woven fabric, mesh

## Abstract

Woven fabric reinforced thermoplastic composites have been gaining significant attention as a lightweight alternative to metal in various industrial fields owing to their high stiffness and strength. Conventional manufacturing processes of woven fabric reinforced thermoplastic composites can be divided into two steps: first, the manufacturing of intermediate material, known as prepreg; then, the formation of the final products from the prepregs. This two-step process increases the manufacturing cost and time of the final composite products. This study demonstrated that woven fabric reinforced thermoplastic composites could be fabricated by an innovative injection molding process instead of the two-step process. A structure placing an extra mesh, which is a new and key component, on the mold-side of woven fabric was devised so that the thermoplastic matrix could be impregnated up to the surface of the woven fabric during injection molding. Tensile tests were performed in the direction parallel to the yarns of the fabric on the injection-molded composites to confirm their mechanical properties. As a result, it was possible to fabricate woven fabric reinforced thermoplastic composites with increased mechanical properties using injection molding without prepreg, and the composites could be molded with a much shorter cycle time than the conventional process, such as thermoforming or over-molding process.

## 1. Introduction

Owing to their high stiffness and strength, fiber-reinforced polymer composites have been gaining significant attention as lightweight alternatives to metals in various industries [1,2,3,4,5], and they are diversely classified depending on their matrices and reinforcements [6,7,8,9,10]. These classifications are outlined in Figure 1. As shown in Figure 1, fiber-reinforced polymer composites can be classified as continuous or discontinuous based on the fiber continuity, and thermoset or thermoplastic composites are based on the type of polymer matrix, etc. [6,8,11,12,13]. Continuous fibers include woven fabric, uni-direction fibers, and braid. Continuous fiber-reinforced composites generally have superior mechanical properties than discontinuous, and thermoplastic composites have better impact resistance than thermoset. A woven fabric reinforced thermoplastic composite is a polymer composite composed of woven fabric and thermoplastic.

The use of thermoplastics as a matrix for polymer composites has several potential advantages [2,12,13,14]. For example, thermoplastics do not require curing reaction; thus, thermoplastic composites can be molded in a shorter cycle time than thermoset composites [15,16]. Thermoplastics are also reusable; thus, thermoplastic composites can be more readily recycled than thermoset composites [17,18]. However, in the use of thermoplastic as a matrix, a significant problem is the difficulty of impregnating fibers with thermoplastic because of the high melting temperature and viscosity of thermoplastics [19,20,21]. Therefore, several studies have been conducted on the use of thermoplastics as matrix for composites [22,23,24,25]. In 1992, Gibson et al. discussed thermoplastic composite molding process models, outlined the key factors influencing the impregnation of carbon and glass fibers with thermoplastic resins, and divided the impregnation process into three types: direct impregnation, intimate mixing prior to melting, and using of low viscosity precursors [26].

The conventional thermoplastic composite molding processes for fabricating a well-impregnated composite can generally be divided into a two-step process [26,27,28,29]. The first step is to manufacture a pre-impregnated intermediate material with thermoplastics, known as prepreg. The next step is to fabricate the final product using the prepreg. Commercially available thermoplastic prepreg manufacturing processes include processes such as pultrusion [30,31,32], extrusion [33,34,35], and double belt press, etc. [36,37,38]. In these processes, a powder form or low-viscosity matrix is used for well-impregnated prepregs [27,39,40]. 

Conventional processes employed to obtain the final product using thermoplastic prepregs include the thermoforming process [41,42,43,44], as shown in Figure 2a, and the over-molding process [45,46] with the thermoformed preform, as shown in Figure 2b. The thermoforming process is a process of forming prepreg into final products or preforms with a high temperature mold. A cycle time of several tens of minutes is required to fabricate the final product because the mold temperature should be higher than the glass transition temperature of the thermoplastic matrix. The prepreg over-molding process is the process of over-molding a preform with a thermoplastic matrix into a final product. A cycle time of several minutes is required because the preform should be preheated prior to inserting into the injection mold. To fabricate woven fabric reinforced thermoplastic composites without prepreg, several studies have been conducted to impregnate dry fibers with thermoplastic directly in resin transfer molding (RTM) [47,48,49] or compression resin transfer molding (CRTM) [50,51,52]. In 2016, Kim et al. assessed the impregnation quality of flax fiber textile reinforced thermoplastic composites manufactured in the CRTM process and compared the process cycle time and mechanical properties of the manufactured final parts with previously reported conventional processes in the literature [51]. In 2019, Studer et al. investigated the potential and mechanism of direct impregnation in the CRTM process using low-viscosity thermoplastics to reduce the process cycle time for automotive thermoplastic composites [52]. In these processes, thermoplastic composites have a moderate impregnation quality, however they can be fabricated in a one-step process with a high-temperature mold.

In the present study, to dramatically reduce the manufacturing cost and cycle time of thermoplastic composites, woven fabric reinforced thermoplastic composites were fabricated using a general injection molding process without a high-temperature mold or prepreg. A new structure was devised to place an extra mesh between the woven fabric and the surface of the mold cavity to impregnate the woven fabric with thermoplastic resin during the injection molding process, as shown in Figure 2c. The most important feature of the new structure is that extra mesh induces a very high-pressure gradient for the impregnation of the thermoplastic matrix. By applying the devised structure, woven fabric reinforced thermoplastic composites can be fabricated by the conventional injection molding process with a shorter cycle time than the general composites molding process.

Thermoplastic injection molding is influenced by several independent process parameters [53,54]. These process parameters, such as mold and melt temperature, injection and packing pressure, and packing and cooling time, affect the shrinkage and deformation of the final product [55,56]. Therefore, several studies have been conducted on the design of algorithms and multi-objective optimization models for the integration of injection molding process parameters [57,58]. In order to improve the quality of the injection-molded product, several studies on a hybrid process with 3D printing technology (direct metal laser sintering and direct-write technology) and injection molding were also conducted [59,60]. In 2022, the effect of barrel temperature, injection pressure, injection speed, and packing time on the characteristics of insert polypropylene (PP) single-polymer composite (SPC) parts was conducted by Wang et al. [61]. The weight and tensile properties of the PP SPC parts varied with the variations of these four parameters in their paper.

As a pre-test, an impregnation experiment was conducted to confirm the degree of impregnation on the surface of the final composite product according to the presence and absence of a mesh. Then, four types of mesh-inserted composites with different fiber volume fractions and mesh-inserted structures were fabricated. Tensile tests were performed to confirm the increase in mechanical properties for the fabricated composites.

## 2. Materials and Experiments

### 2.1. Materials and Methods

Three types of materials were used in this experiment: the thermoplastic matrix, carbon woven fabric, and aluminum mesh. The polypropylene (J-150, Lotte chemical) used in general injection molding was chosen as the matrix, the properties of which are shown in Table 1.

As shown in Figure 2c, carbon woven fabrics and extra meshes were used as reinforcements. Six types of mesh were tested, and only the specifications of the aluminum mesh showing the best results with the woven fabric used are shown in Table 2. The schematics of the woven fabric and mesh are shown in Figure 3. As shown in Figure 3, woven fabric and mesh are plain woven, and both the woven fabric and mesh have pores made by weft and warp. In this article, the woven fabric and mesh are represented by simplified symbols, as shown in Figure 3a,b. The top view images and geometric information of two woven fabrics and a mesh are shown in Figure 4 and Table 2. As shown in Figure 3 and Figure 4, the yarns of carbon fabric are composed of thousands of fiber filaments. The porosity of the fabrics and mesh was defined as the ratio of the area of pores per unit area. Fabric A (1k) and Fabric B (3k) have different densities, types of yarns, and porosities; the porosities of the woven fabrics are much lower than the porosity of the mesh.

The LGE110 injection molding machine (LS Mtron) was used as shown in Figure 5a, and the specification of the injection molding machine is shown in Table 3. The mold has two cavities on the fixed and moving sides, respectively, as shown in Figure 5b. The cavities have the shape of a rectangle (50 mm × 100 mm) with a depth of 1.50 mm on both sides.

The surface of the injection-molded composite was sputtered with gold and then observed through scanning electron microscopy (SEM, HITACHI SU8020). The mesh-inserted composites were cut into dog bone-shaped specimens (ISO 527) using a water jet. The tensile properties of specimens were measured using a Zwick Z010TE universal testing machine. A 10 kN load cell was used and the crosshead speed was 2.0 mm/min. At least five tensile specimens for each structure were tested at room temperature (23 °C).

### 2.2. Experiment

#### 2.2.1. Impregnation Experiment

To fabricate woven fabric reinforced thermoplastic using a conventional injection molding process, a sufficient pressure gradient is required to enable the molten resin to be impregnated into the dense woven fabrics placed in the mold cavities. However, when only the woven fabrics are placed on the surface of mold cavities, as shown in Figure 6(a1), the surface of the composite cannot be impregnated with the resin, as shown in Figure 6(a2). The reason for this is that a very high pressure drop occurs at the pores of woven fabric, as shown in Figure 6(a3). This is very similar to the short shot error in the injection molding field.

Therefore, a new structure was devised in which a mesh was placed between the woven fabric and the surface of the mold cavity, so that the surface of the composite could be impregnated with the molten resin, as shown in Figure 6(b1). When the mesh is placed between the woven fabric and the surface of the mold cavity, the surface of composite can be impregnated with the resin, as shown in Figure 6(b2). This is because a sufficient pressure gradient could be generated such that the molten resin can penetrate the pores of the woven fabric, as shown in Figure 6(b3).

The SEM images on the surface of the cavity side for the injection-molded composites are shown in Figure 6. The surface of the injection-molded composite without a mesh is not impregnated with resin, and the yarns of woven fabric are exposed, as shown in Figure 6(a4). However, the surface of injection-molded composites with meshes are covered with resin because of the penetration of the molten resin at the pores of the woven fabric, as shown in Figure 6(b4). These samples were molded in the same molding conditions: the mold temperature (50 °C), melt temperature (240 °C), injection time (1.0 s), packing pressure (30.0 MPa), packing time (2.0 s), and the cooling time (50.0 s). As a result of the impregnation experiment, when the proper size of meshes were placed between the woven fabric and the surface of the mold, the surface of the final sample could be sufficiently impregnated with the resin while the conventional injection molding process was used.

#### 2.2.2. Injection Molding Experiment of Mesh-Inserted Composite

Based on the results of the impregnation experiment, the injection molding experiments of mesh-inserted composites were performed to demonstrate the feasibility of fabricating woven fabric reinforced thermoplastic composites. Four types of composites with a different fiber volume fraction and inserted structures were fabricated in the injection molding experiment, as shown in Figure 7.

2-MF structure is a structure in which a woven fabric and a mesh are alternately laminated in two layers, as outlined in Figure 7a, and 4-MF structure is a structure in which the woven fabrics and the meshes are alternately laminated in four layers, as outlined in Figure 7b. Two types of woven fabric, Fabric A (1k) and Fabric B (3k), were used and laminated into two types of structures, 2-MF and 4-MF, respectively. The woven fabrics and meshes of the 2-MF and 4-MF structures were placed on the surface of the moving and fixed sides of the mold cavity, respectively, as shown in Figure 7c, and were then injection molded. The injection molding conditions are outlined in Table 4, the mesh-inserted composite could be molded in less than 1 min for most of applications, including the cooling time.

The composition of the four types of molded composites with different woven fabrics and laminated structure are shown in Table 5. Cases 1 and 2, and cases 3 and 4 have the same 2-MF and 4-MF structures, respectively. The calculated fiber volume fractions of samples shown in Table 5 are different due to the different types of woven fabrics. The optical microscopy (RAM Optical Instrumentation, Datastar 200) was performed to confirm the alignment and arrangement of the fabrics and meshes inside the injection-molded composites. The mesh-inserted composites were bonded with epoxy, and the cross-section of composites was polished. When the porosity of the mesh is too low, the molten resin cannot sufficiently penetrate through the pores in the fabric. On the other hand, when the porosity of the mesh is too high, sagging of the fabric can occur at the pores of the mesh. The optical microscopy of the cross-section of the 2-MF and 4-MF structure composites with aluminum meshes are shown in Figure 8. In case of using the aluminum mesh among the six types of mesh, the woven fabrics placed between the meshes could be aligned relatively well without wrinkles or sagging. These images show the structure when the molten resin penetrates the pores of the woven fabric successfully and fills the additional space secured by the mesh between the woven fabric and surface of the mold.

To confirm that the fiber filaments of woven fabric were impregnated with the molten resin, SEM observations were performed on the warp and weft of the 4-MF (3k) composite (case #4), which had the highest fiber volume fraction among the molded composites. The SEM images of the 4-MF composite are shown in Figure 9. These images show that the molten resin not only filled in the pores of woven fabric, but also covered the fiber filaments of the yarns.

## 3. Results and Discussion

Tensile tests were performed in the direction parallel to the yarns of the woven fabric to confirm the mechanical properties of the mesh-inserted composites, as shown in Figure 10a,b. Table 6 shows the comparison of the tensile test results of the virgin PP and the mesh-inserted composites. The respective tensile stress–strain curves are shown in Figure 10c.

The 2-MF (1k) composite (case #1) with a volume fraction of 4.0% had a tensile modulus of 3.58 GPa, which was greater than that of the virgin matrix by a factor of approximately 1.97, and a maximum tensile strength of 54.0 MPa, which was greater than that of the virgin matrix by a factor of approximately 1.69. The 2-MF (1k) composites and the 2-MF (3k) composites (case #1 and case #2) have the same 2-MF structure, however the 2-MF (3k) composites have a higher fiber volume fraction than 2-MF (1k) composites. The 2-MF (3k) composites, with a volume fraction of 8.4%, had a tensile modulus of 4.35 GPa and a maximum tensile strength of 72.0 MPa. The 2-MF (3k) composites and the 4-MF (1k) composites (case #2 and case #3) have a similar volume fraction, however the types of structure are different. The 4-MF (1k) composites with a volume fraction of 7.9% had a tensile modulus of 6.63 GPa, which is greater than the tensile modulus of 2-MF (3k) composites. On the other hand, the maximum tensile strength of 2-MF (3k) composites and 4-MF (1k) composites is similar. The 4-MF (3k) composite (case #4) with a fiber volume fraction of 16.2% had a tensile modulus of 7.16 GPa, greater than that of the virgin matrix by a factor of approximately 4.18, and a maximum tensile strength of 108.4 MPa, greater than that of the virgin matrix by a factor of approximately 3.39. According to the tensile tests, both the tensile modulus and maximum tensile strength of the mesh-inserted composites increase with an increase in the fiber volume fraction and number of laminated layers of woven fabrics and meshes, generally.

The mechanical properties of mesh-inserted woven fabric reinforced thermoplastic composites can be compared with the mechanical properties of injection-molded short fiber reinforced thermoplastic composites (SFRTP) and woven fabric reinforced thermoplastic composites (WFRTP) made by thermoforming. Even though the tensile modulus can reach very high levels, the tensile strength of injection-molded SFRTPs has a limitation of about 2.0 for that of the virgin matrix, even if the fiber volume fraction is increased [62]. Furthermore, injection-molded SFRTPs have a significant difference of mechanical properties in the machine direction (MD) and transverse direction (TD) to the flow of resin due to the anisotropy of fibers [63]. Thermoformed WFRTPs, which are composed continuous fibers as the reinforcement, can improve these disadvantages of SFRTPs; however, they usually require tens of minutes of cycle time, as the extensive heating and cooling time of the mold is essential [44,64]. The mesh-inserted woven fabric reinforced thermoplastic composites made by injection molding can improve the disadvantages of conventional SFRTPs by using the woven fabrics as the reinforcements, and can be fabricated in a much shorter cycle time than WFRTPs because they are molded by a conventional injection molding process.

## 4. Conclusions

The woven fabric reinforced thermoplastic composites were fabricated using a conventional injection molding process instead of the thermoforming or prepreg over-molding. A new structure was devised in which a mesh was placed between the woven fabric and the surface of the mold. This unique structure facilities the advantageous impregnation of thermoplastic resin into the woven fabric. The final mesh-inserted composites with a fiber volume fraction of 4.0~16.2% were fabricated in the present study. Tensile tests were performed on the mesh-inserted woven fabric reinforced thermoplastic composites in the direction parallel to the yarns of the woven fabric. Depending on the types of structure, it was confirmed that a tensile modulus and tensile strength could be controlled to increase up to 4.18 and 3.39 times higher than those of the virgin matrix, respectively. Therefore, the types of woven fabrics, the number and material of meshes, and the matrix can be selected for a type of structure required for the mechanical properties needed. In other words, this innovative injection molding process has the potential to fabricate thermoplastic composites with various configurations depending on the purpose and mechanical properties, i.e., automotive parts or drone bodies. Additional studies will be conducted on the mechanical properties of mesh-inserted composites depending on the direction of woven fabrics and meshes placed in the mold cavity.

## Figures and Tables

**Figure 1 polymers-14-01577-f001:**
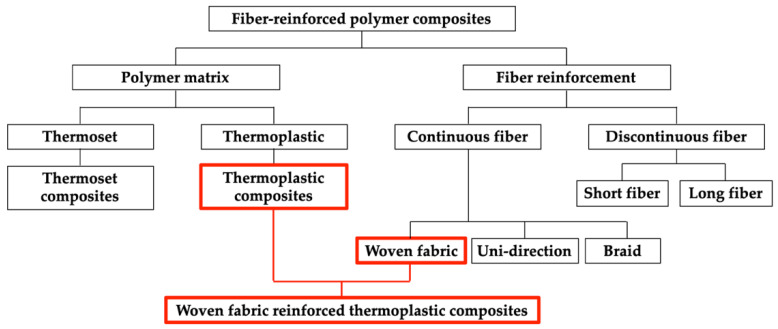
Classification of fiber-reinforced polymer composites.

**Figure 2 polymers-14-01577-f002:**
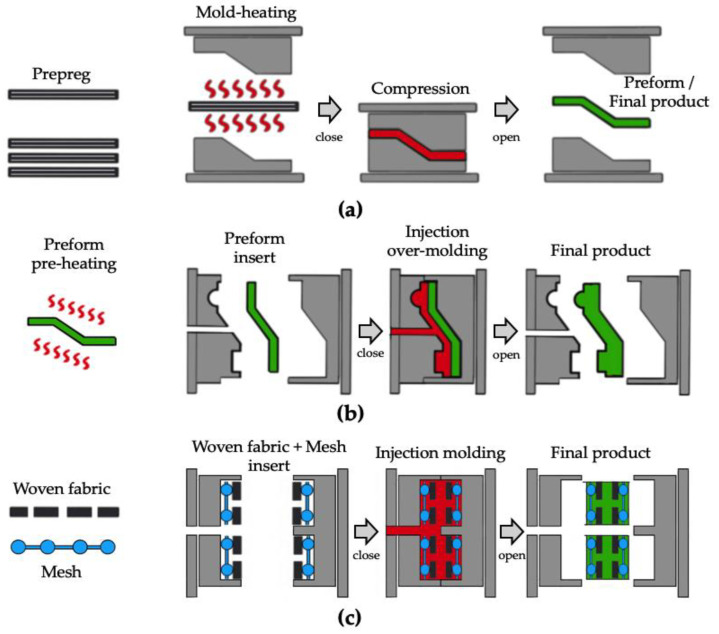
Comparison of woven fabric reinforced thermoplastic composites molding process: (**a**) press thermoforming; (**b**) prepreg injection over-molding; (**c**) mesh-inserted injection molding.

**Figure 3 polymers-14-01577-f003:**
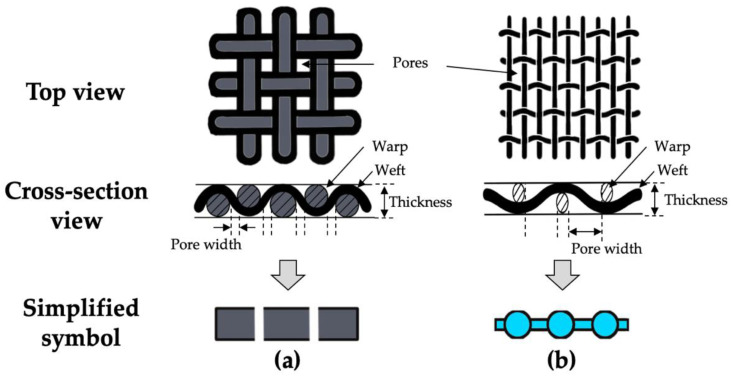
Schematics of the top view, cross-section view, and simplified symbol: (**a**) woven fabric; (**b**) mesh.

**Figure 4 polymers-14-01577-f004:**
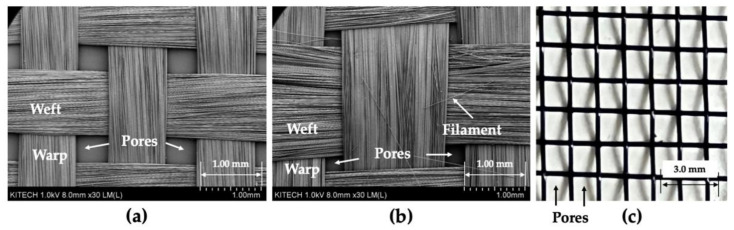
Images of the woven fabric and mesh: (**a**) fabric A (1k); (**b**) fabric B (3k); (**c**) mesh.

**Figure 5 polymers-14-01577-f005:**
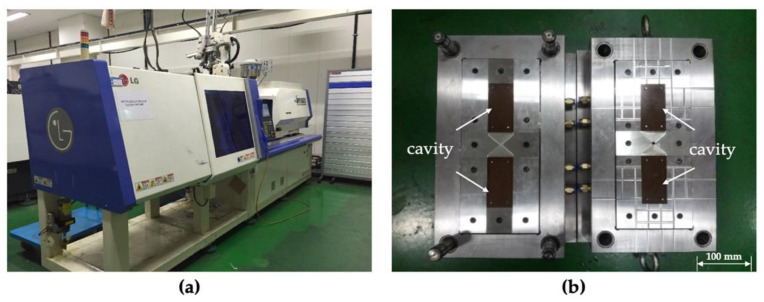
Photographs of the (**a**) injection molding machine and (**b**) mold used.

**Figure 6 polymers-14-01577-f006:**
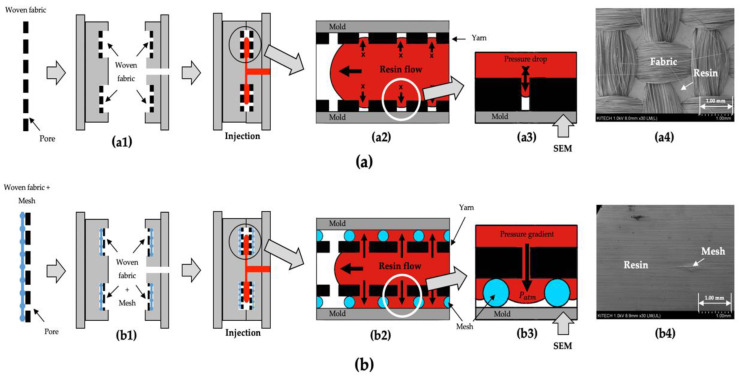
Schematics of the impregnation experiment and SEM images on the surface of cavity side for injection-molded composites: (**a**) placing only the woven fabrics; (**b**) placing meshes between the woven fabric and mold surface.

**Figure 7 polymers-14-01577-f007:**
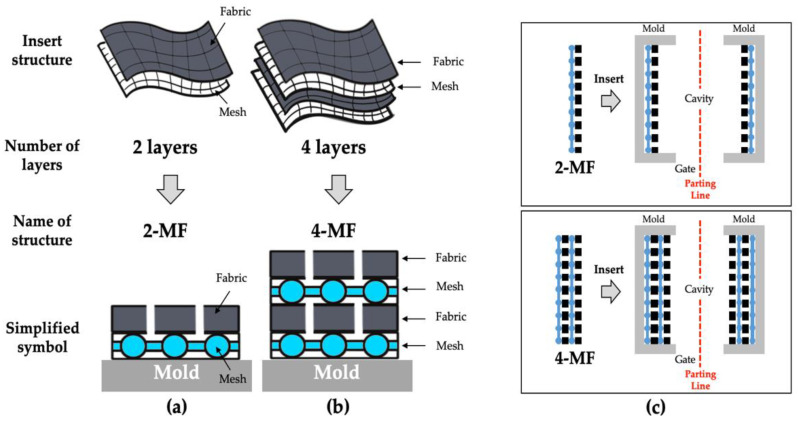
Schematics of the insert structures: (**a**) 2-MF structure; (**b**) 4-MF structure; (**c**) mold insert structure.

**Figure 8 polymers-14-01577-f008:**
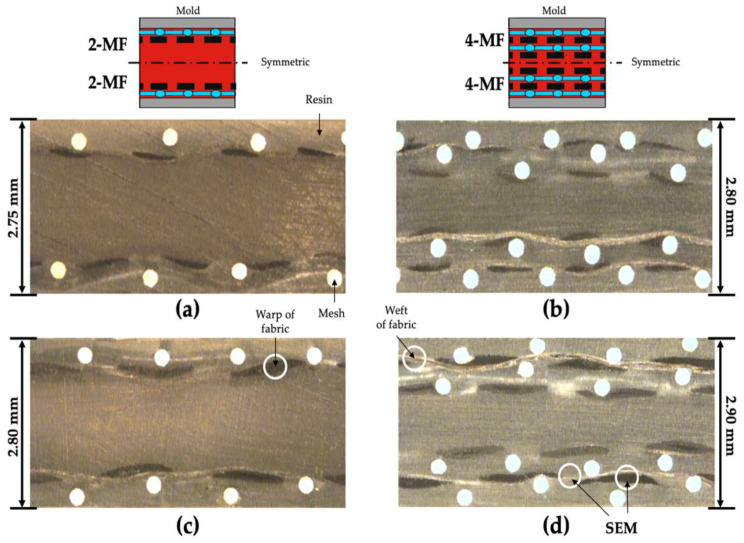
Optical microscopy images of the cross-section of the molded composites: (**a**) 2-MF (1k); (**b**) 4-MF (1k); (**c**) 2-MF (3k); (**d**) 4-MF (3k).

**Figure 9 polymers-14-01577-f009:**
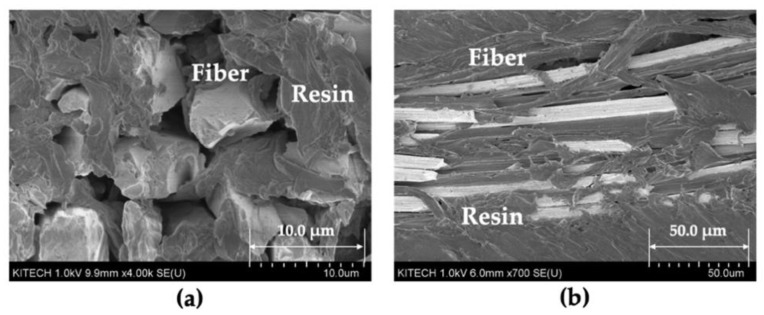
SEM images of the 4-MF (3k) composites: (**a**) cross-section of warp; (**b**) cross-section of weft.

**Figure 10 polymers-14-01577-f010:**
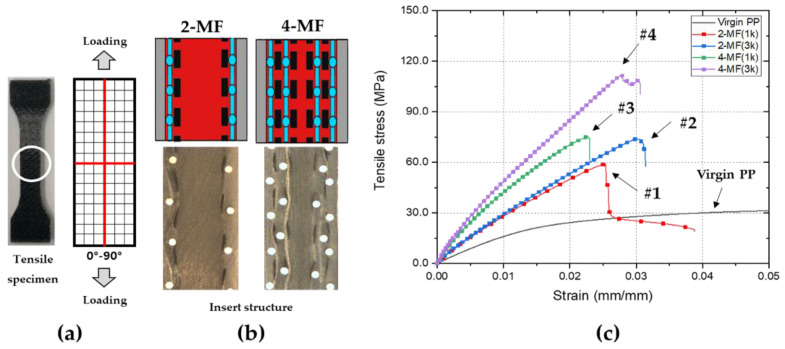
Results of the tensile tests of mesh-inserted composites: (**a**) direction of loading; (**b**) cross-section images of composites; (**c**) stress–strain curves of composites and virgin matrix.

**Table 1 polymers-14-01577-t001:** Properties of PP (J-150).

Property	Method	Value
Density	ASTM D1505 ^1^	0.90 g/cm^3 1^
Melt index (230 °C, 2.16 kg)	ASTM D1238 ^1^	10 g/10 min ^1^
Tensile modulus	ISO 527 ^2^	1.82 GPa ^2^
Tensile strength	ISO 527 ^2^	32.0 MPa ^2^
Heat deflection temperature (0.46 MPa)	ASTM D648 ^1^	119.0 °C ^1^

^1^ Lotte chemical data sheet. ^2^ KITECH experimental results.

**Table 2 polymers-14-01577-t002:** Geometric information of the woven fabrics and mesh.

Type	Types of Yarn	Thickness (mm)	Density (Counts/in)	Porosity	Weight (g/m^2^)
Fabric A	Carbon 1k	0.14	17.5	0.10	95
Fabric B	Carbon 3k	0.27	13.0	0.06	208
Mesh	Aluminum (Al)	0.50	18.0	0.65	174

**Table 3 polymers-14-01577-t003:** Specification of injection molding machine (LGE110).

Item	Value
Screw diameter (mm)	25.0
Maximum injection stroke (mm)	111.0
Injection capacity (cm^3^)	185.0
Maximum injection speed (mm/s)	350.0
Maximum injection pressure (kgf/cm^2^)	2600
Maximum clamping force (ton)	110

**Table 4 polymers-14-01577-t004:** Injection molding conditions for the mesh-inserted composite.

Process Condition	Value
Melt temperature (°C)	240
Mold temperature (°C)	50
Injection time (s)	1.0
Injection Speed (mm/s)	45.0
Back pressure (MPa)	5.0
V/P switch over pressure (MPa)	35.0
Packing pressure (MPa)	30.0
Packing time (s)	2.0
Cooling time (s)	50
Total cycle time	Less than 1 min

**Table 5 polymers-14-01577-t005:** Compositions of mesh-inserted composites.

Type of Structure	Case Number	Name of Composite	Type of Fabric	Fiber Volume Fraction (%)	Mesh Volume Fraction (%)
2-MF	#1	2-MF (1k)	Fabric A (1k)	4.0	4.9
	#2	2-MF (3k)	Fabric B (3k)	8.4	4.7
4-MF	#3	4-MF (1k)	Fabric A (1k)	7.9	9.4
	#4	4-MF (3k)	Fabric B (3k)	16.2	9.2

**Table 6 polymers-14-01577-t006:** Comparison of tensile test results.

Type of Structure	Case Number	Fiber Volume Fraction (%)	Name of Composite	Tensile Modulus (GPa) (%/Virgin)	Tensile Strength (MPa) (%/Virgin)
Matrix	-	-	Virgin PP	1.82 ± 0.04	32.0 ± 0.3
2-MF	#1	4.0	2-MF (1k)	3.58 ± 0.11 (197%)	54.0 ± 3.3 (169%)
	#2	8.4	2-MF (3k)	4.35 ± 0.08 (239%)	72.0 ± 1.6 (225%)
4-MF	#3	7.9	4-MF (1k)	6.63 ± 0.18 (364%)	68.3 ± 5.5 (213%)
	#4	16.2	4-MF (3k)	7.61 ± 0.14 (418%)	108.4 ± 5.3 (339%)

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
