# Peer review of "Innovative Injection Molding Process for the Fabrication of Woven Fabric Reinforced Thermoplastic Composites"

_polymers, 2022, doi:10.3390/polym14081577_

Round 1

Reviewer 1 Report

The authors have demonstrated a unique single step manufacturing operation where a weave + mesh structure is placed in the mold and injection molding is used to impregnate the woven fabric thereby having better cycle times compared to thermoforming or over-molding. Furthermore, tensile tests showed the use of the weave + mesh structure increased tensile strength and modulus upto 4.18 and 3.39 times virgin matrix material.  

While the process and components (mesh and woven fabric) seem unique and interesting the authors would need to address the following points in order for this manuscript to be accepted to this prestigious journal:

  1. Figure 2b: Are the authors sure that in this process the mold is opened after the preform is formed? As per the reviewer’s knowledge the mold remains closed after forming and overmolding takes place (https://www.youtube.com/watch?v=HJ_N555hWTs).
  2. Figure 2. Cycle times need to be prescribed for each of these processes.
  3. No mention of the equipment that was used to test tensile specimens like load frame, load cell, strain rate, and standard used.
  4. Table 3. replace resin temperature with matrix temperature. 
  5. The experiments section is missing details on SEM imaging and Optical Imaging (what equipment, were samples sputter coated ?). Similarly, details on the IM machine and its technical specification need to be elaborate.
  6. The authors need to mention the process parameters on the Injection Molding side for initial experiments. Also, more process parameters need to be reported like injection pressure, back pressure, injection speed, overall cycle time
  7. Table 5/Figure 10: Was only 1 sample tested per formulation? If so that the result is void and the tests need to be repeated in order to obtain a statistically sound result. If multiple samples were carried out then average and standard deviation need to be reported !
  8. Figure 10 shows dogbone shaped tensile coupouns while the cavity is a rectangular plaque. Where they waterjet or punched ? The authors need to provide more details on how these tensile samples were prepared in the experimental section.
  9. The authors mention: The mechanical properties of mesh-inserted woven fabric reinforced thermoplastic 228 composites can be compared with the mechanical properties of injection-molded short 229 fiber reinforced thermoplastic composites (SFRTP) and woven fabric reinforced thermo-230 plastic composites (WFRTP) made by thermoforming. Did the authors test this ? If yes provide numbers ? If not provide a reference.
  10. Line 67 compression resin transfer molding (CTRM) correct abbreviation is CRTM
  11. Is there any way for the authors to quantify impregnation between he 4 cases ? Or can NDE measurements be undertaken to ascertain void content ? This information may be very fruitful for this paper and readers in general. 
  12. Have the authors investigated the crystallinity of the different composite cases and virgin matrix ? This might be worthwhile as well. 
  13. The authors should cite the following paper in the introduction-

    Effects of Injection Molding Parameters on Properties of Insert-Injection Molded Polypropylene Single-Polymer Composites

Author Response

Dear Reviewer,

Thank you for your comment.

We did our best to write the revision.

"The revision is attached."

I'll be looking forward to your good reply.

Best regards.

Response to Reviewer 1 Comments

Point 1: Figure 2b: Are the authors sure that in this process the mold is opened after the preform is formed? As per the reviewer’s knowledge the mold remains closed after forming and overmolding takes place (https://www.youtube.com/watch?v=HJ_N555hWTs).

=> Response 1: Of course, the mold remains closed during overmolding stage. However, the preform is usually molded by separate process (pultrusion or press molding) and inserted into the final mold. Once again the mold remains closed during preform molding stage. The figure right shows the final product after ejection. We made minor change in Fig. 2(b) as your comment.

Point 2: Figure 2. Cycle times need to be prescribed for each of these processes.

=> Response 2: Extra informations about cycle times for each process were added in Lines [64-71], Lines [89-91].

Point 3: No mention of the equipment that was used to test tensile specimens like load frame, load cell, strain rate, and standard used.

=> Response 3: Extra information for tensile tests was added in Lines [227-229].

Point 4: Table 3. replace resin temperature with matrix temperature.

=> Response 4: We replaced “Resin temperature” to “Melt(matrix) temperature” in Table 4. In injection molding field we used to call “Melt(matrix) temperature” instead of “Resin temperature”. Thank you for pointing out our mistake.

Point 5: The experiments section is missing details on SEM imaging and Optical Imaging (what equipment, were samples sputter coated ?). Similarly, details on the IM machine and its technical specification need to be elaborate.

=> Response 5: Extra information for SEM imaging was added in Lines [161-162]. Extra information for Optical imaging was added in Line [198]. Detailed information for IM machine and its technical specification were added in the Table 3 and Line [134].

Point 6: The authors need to mention the process parameters on the Injection Molding side for initial experiments.

=> Response 6: Process conditions for initial experiments were added in Lines [167-170].

Also, more process parameters need to be reported like injection pressure, back pressure, injection speed, overall cycle time.

=> Response 6: Information of more process conditions for the experiments were added in Table 4 (Line [192]).

Point 7: Table 5/Figure 10: Was only 1 sample tested per formulation? If so that the result is void and the tests need to be repeated in order to obtain a statistically sound result. If multiple samples were carried out then average and standard deviation need to be reported !

=> Response 7: Detailed information about number of sampes tested and its standard deviation were added in Table 6 and Lines [229-230, 233].

Point 8: Figure 10 shows dogbone shaped tensile coupouns while the cavity is a rectangular plaque. Where they waterjet or punched ? The authors need to provide more details on how these tensile samples were prepared in the experimental section.

=> Response 8: Extra information for tensile tests specimens was added in Lines [226-227].

Point 9: The authors mention: The mechanical properties of mesh-inserted woven fabric reinforced thermoplastic 228 composites can be compared with the mechanical properties of injection-molded short 229 fiber reinforced thermoplastic composites (SFRTP) and woven fabric reinforced thermo-230 plastic composites (WFRTP) made by thermoforming. Did the authors test this ? If yes provide numbers ? If not provide a reference.

=> Response 9: The mechanical properties of SFRTP were from Ref. #55 & #56. And the mechanical properties of WFRTP were from Ref. #43 & #54.

Point 10: Line 67 compression resin transfer molding (CTRM) correct abbreviation is CRTM.

=> Response 10: We changed “CTRM” to “CRTM” in Line [73]. Thank you for your correction.

Point 11: Is there any way for the authors to quantify impregnation between the 4 cases ? Or can NDE measurements be undertaken to ascertain void content ? This information may be very fruitful for this paper and readers in general.

=> Response 11: In this paper, direct optical imaging measurements were taken as shown in Figure 8. We will consider Nondestructed Evaluation (NDE) for our future study as a basis of your recommendation.

Point 12: Have the authors investigated the crystallinity of the different composite cases and virgin matrix ? This might be worthwhile as well.

=> Response 12: In this paper, we paid attention to the macroscopic tensile properties rather than microscopic structure such as crystalinity. We may consider the effect of crystalinity on the final quality of the product in our future study.

Point 13: The authors should cite the following paper in the introduction-Effects of Injection Molding Parameters on Properties of Insert-Injection Molded Polypropylene Single-Polymer Composites.

=> Response 13: The reference paper you mentioned was added and cited in Lines [96-100]. Thank you for recommendation.

Reviewer 2 Report

Paper ID: polymers-1658103

  1. This study proposed an innovative injection molding process for the fabrication of woven fabric reinforced thermoplastic composites. It seems interesting. Congratulations for a useful research publication. Manuscript is well written and structured. Literature is well reviewed. Research data is available for reproducing experiments and procedures. Proper characterization techniques were used. Results are conveniently analyzed and commented.
  2. Figure 5 (b)—Please mark the scale bar.
  3. Figure 10 (c)---Please amend it since it is not clear.
  4. Some leading works regarding injection molding should be discussed in the introduction.
    ---Kitayama, S., Hashimoto, S., Takano, M. et al. Multi-objective optimization for minimizing weldline and cycle time using variable injection velocity and variable pressure profile in plastic injection molding. Int J Adv Manuf Technol 107, 3351–3361 (2020).
    ---Zhao, Ny., Lian, Jy., Wang, Pf. et al. Recent progress in minimizing the warpage and shrinkage deformations by the optimization of process parameters in plastic injection molding: a review. Int J Adv Manuf Technol (2022). https://doi.org/10.1007/s00170-022-08859-0
    ---Alvarado-Iniesta, A., Cuate, O. & Schütze, O. Multi-objective and many objective design of plastic injection molding process. Int J Adv Manuf Technol 102, 3165–3180 (2019).
    ---Zhou, H., Zhang, S. & Wang, Z. Multi-objective optimization of process parameters in plastic injection molding using a differential sensitivity fusion method. Int J Adv Manuf Technol 114, 423–449 (2021).
    ---Kuo, CC., Yang, XY. Optimization of direct metal printing process parameters for plastic injection mold with both gas permeability and mechanical properties using design of experiments approach. Int J Adv Manuf Technol 109, 1219–1235 (2020).
    ---Loaldi, D.; Piccolo, L.; Brown, E.; Tosello, G.; Shemelya, C.; Masato, D. Hybrid Process Chain for the Integration of Direct Ink Writing and Polymer Injection Molding. Micromachines2020, 11, 509.
    ---Carrupt, M.C.; Piedade, A.P. Modification of the Cavity of Plastic Injection Molds: A Brief Review of Materials and Influence on the Cooling Rates. Materials 2021, 14, 7249. 
    ---Giang, N.T.; Minh, P.S.; Son, T.A.; Uyen, T.M.T.; Nguyen, T.-H.; Dang, H.-S. Study on External Gas-Assisted Mold Temperature Control with the Assistance of a Flow Focusing Device in the Injection Molding Process. Materials 2021, 14,
    ---Kusić, D.; Božič, U.; Monzón, M.; Paz, R.; Bordón, P. Thermal and Mechanical Characterization of Banana Fiber Reinforced Composites for Its Application in Injection Molding. Materials 2020, 13, 3581.
  5. What's the novelty of the research in the paper? Please clarify.
  6. Authors should more clearly emphasize the contribution of this work in relation to the existing solutions in the literature.
  7. What is the main difficulty when applying proposed method? Authors should clearly state the limitations of the proposed method in practical applications.
  8. Please show some directions for the future study.
  9. The abstract is also not sufficiently informative, concise and clear. No any quantitative results. Please amend it.
  • Conclusions must be comprehensive and not written like a report. Please amend it.
  • Please add the applicability of present work in the conclusion section.
  • Does your paper have industrial applications? If yes, who are the likely users?
  • What is your main contribution to the field?
  • Please explain more regarding Figures 1, 3, 5, 7, and 8.
  • Finally, I would suggest the author to address the questions above in the revision. I am pleased to review the revised manuscript.

Author Response

Dear Reviewer,

Thank you for your comment.

We did our best to write the revision.

"The revision is attached."

I'll be looking forward to your good reply.

Best regards.

Response to Reviewer 2 Comments

Point 1 => Overview

Point 2: Figure 5 (b)—Please mark the scale bar.

=> Response 2: We added the scale bar in Figure 5(b).

Point 3: Figure 10 (c)---Please amend it since it is not clear.

=> Response 3: We made the symbols and marks more clear in Figure 10(c).

Point 4: Some leading works regarding injection molding should be discussed in the introduction.

=> Response 4: The reference papers you mentioned were added and cited in Lines [95-96]. Thank you for recommentadtion.

Point 5: What's the novelty of the research in the paper? Please clarify.

=> Response 5: We have shown a unique single step manufacturing process of TPC (Thermoplastic composite) using a woven fabric + mesh structure and conventional injection molding. No one tried this unique strucre before. (Lines [82-89].)

Point 6: Authors should more clearly emphasize the contribution of this work in relation to the existing solutions in the literature.

=> Response 6: Extra explanation for the contribution of this work was added in Lines [89-91].

Point 7: What is the main difficulty when applying proposed method? Authors should clearly state the limitations of the proposed method in practical applications.

=> Response 7: Our proposed composite structure has the limitation to make the final product to be isotropic. More studies are undertaken to test various combination of placing woven fabrics in different direction. (Lines [287-289])

Point 8: Please show some directions for the future study.

=> Response 8: As mentioned above more studies are undertaken to test various combination of placing woven fabrics in different directions. (Lines [287-289])

Point 9: The abstract is also not sufficiently informative, concise and clear. No any quantitative results. Please amend it.

=> Response 9: Because our quantitative result are described in conclusions again, we deleted all the quantitative things and revised the abstract concise.

Point 10-1: Conclusions must be comprehensive and not written like a report. Please amend it.

=> Response 10-1: More explanation for the idea of this work was added in Lines [275-728]. And we made the conclusion comprehensive.

Point 10-2: Please add the applicability of present work in the conclusion section. Does your paper have industrial applications? If yes, who are the likely users?

=> Response 10-2: Extra explanation for the application was added in Line [287].

Point 10-3: What is your main contribution to the field?

=> Response 10-3: Depends on the design limitation of mechanical strength or modulus we can provide required structure of the final composite product in advance.

Point 10-4: Please explain more regarding Figures 1, 3, 5, 7, and 8.

=> Response 10-4: Extra explanation regarding Figure 1, 3, and 5 were added in Lines [34-37, 118-121, 134-137].

Round 2

Reviewer 1 Report

Dear Authors

Shift lines 226 to 230 to materials and methods. 

Shift lines 161 to 162 to materials and methods. 

Table 4: just melt temperature is fine. Don't mention matrix

Everything else checks out. I recommend this paper for publishing.  

Author Response

Dear Reviewer,

We devised the revision (Round 2) based on your fruitful comments.

"The revision (Round 2) is attached."

Best regards.

Response to Reviewer 1 Comments

Point 1: Shift lines 226 to 230 to materials and methods. 

=> Response 1: The lines [226-230] in the revised file (Round 1) were shifted to materials and methods section, lines[147-151] in the revised file (Round 2).

Point 2: Shift lines 161 to 162 to materials and methods. 

=> Response 2: The lines [161-162] in the revised file (Round 1) were shifted to materials and methods, lines[146-147] in the revised file (Round 2).

Reviewer 2 Report

Even after the revision, the quality of the paper is not improved.

  1. Some leading works regarding injection molding should be discussed in the introduction.
    ---Kitayama, S., Hashimoto, S., Takano, M. et al. Multi-objective optimization for minimizing weldline and cycle time using variable injection velocity and variable pressure profile in plastic injection molding. Int J Adv Manuf Technol 107, 3351–3361 (2020).
    ---Zhao, Ny., Lian, Jy., Wang, Pf. et al. Recent progress in minimizing the warpage and shrinkage deformations by the optimization of process parameters in plastic injection molding: a review. Int J Adv Manuf Technol (2022). https://doi.org/10.1007/s00170-022-08859-0
    ---Alvarado-Iniesta, A., Cuate, O. & Schütze, O. Multi-objective and many objective design of plastic injection molding process. Int J Adv Manuf Technol 102, 3165–3180 (2019).
    ---Zhou, H., Zhang, S. & Wang, Z. Multi-objective optimization of process parameters in plastic injection molding using a differential sensitivity fusion method. Int J Adv Manuf Technol 114, 423–449 (2021).
    ---Kuo, CC., Yang, XY. Optimization of direct metal printing process parameters for plastic injection mold with both gas permeability and mechanical properties using design of experiments approach. Int J Adv Manuf Technol 109, 1219–1235 (2020).
    ---Loaldi, D.; Piccolo, L.; Brown, E.; Tosello, G.; Shemelya, C.; Masato, D. Hybrid Process Chain for the Integration of Direct Ink Writing and Polymer Injection Molding. Micromachines2020, 11, 509.
    ---Carrupt, M.C.; Piedade, A.P. Modification of the Cavity of Plastic Injection Molds: A Brief Review of Materials and Influence on the Cooling Rates. Materials 2021, 14, 7249. 
    ---Giang, N.T.; Minh, P.S.; Son, T.A.; Uyen, T.M.T.; Nguyen, T.-H.; Dang, H.-S. Study on External Gas-Assisted Mold Temperature Control with the Assistance of a Flow Focusing Device in the Injection Molding Process. Materials 2021, 14,
    ---Kusić, D.; Božič, U.; Monzón, M.; Paz, R.; Bordón, P. Thermal and Mechanical Characterization of Banana Fiber Reinforced Composites for Its Application in Injection Molding. Materials 2020, 13, 3581.

Author Response

Dear Reviewer,

Thank you for recommending the reference papers.

The all reference papers you mentioned were added and cited to improve the quality of the paper.

"The revision (Round 2) is attached."

Best regards.

Response to Reviewer 2 Comments

Point 1: Some leading works regarding injection molding should be discussed in the introduction.

=> Response 1: The all reference papers you mentioned were added and cited as follows:

---Kitayama, S., Hashimoto, S., Takano, M. et al. Multi-objective optimization for minimizing weldline and cycle time using variable injection velocity and variable pressure profile in plastic injection molding. Int J Adv Manuf Technol 107, 3351–3361 (2020).

=> This paper is added and cited in Lines [95-96, Ref #53].

---Zhao, Ny., Lian, Jy., Wang, Pf. et al. Recent progress in minimizing the warpage and shrinkage deformations by the optimization of process parameters in plastic injection molding: a review. Int J Adv Manuf Technol (2022). https://doi.org/10.1007/s00170-022-08859-0

=> This paper is added and cited in Lines [96-98, Ref #55].

---Alvarado-Iniesta, A., Cuate, O. & Schütze, O. Multi-objective and many objective design of plastic injection molding process. Int J Adv Manuf Technol 102, 3165–3180 (2019).

=> This paper is added and cited in Lines [98-100, Ref #57].

---Zhou, H., Zhang, S. & Wang, Z. Multi-objective optimization of process parameters in plastic injection molding using a differential sensitivity fusion method. Int J Adv Manuf Technol 114, 423–449 (2021).

=> This paper is added and cited in Lines [98-100, Ref #58].

---Kuo, CC., Yang, XY. Optimization of direct metal printing process parameters for plastic injection mold with both gas permeability and mechanical properties using design of experiments approach. Int J Adv Manuf Technol 109, 1219–1235 (2020).

=> This paper is added and cited in Lines [100-102, Ref #59].

---Loaldi, D.; Piccolo, L.; Brown, E.; Tosello, G.; Shemelya, C.; Masato, D. Hybrid Process Chain for the Integration of Direct Ink Writing and Polymer Injection Molding. Micromachines2020, 11, 509.

=> This paper is added and cited in Lines [100-102, Ref #60].

---Carrupt, M.C.; Piedade, A.P. Modification of the Cavity of Plastic Injection Molds: A Brief Review of Materials and Influence on the Cooling Rates. Materials 2021, 14, 7249. 

=> This paper is added and cited in Lines [96-98, Ref #56].

---Giang, N.T.; Minh, P.S.; Son, T.A.; Uyen, T.M.T.; Nguyen, T.-H.; Dang, H.-S. Study on External Gas-Assisted Mold Temperature Control with the Assistance of a Flow Focusing Device in the Injection Molding Process. Materials 2021, 14,

=> This paper is added and cited in Lines [95-96, Ref #54].

---Kusić, D.; Božič, U.; Monzón, M.; Paz, R.; Bordón, P. Thermal and Mechanical Characterization of Banana Fiber Reinforced Composites for Its Application in Injection Molding. Materials 2020, 13, 3581.

=> This paper is added and cited in Lines [28-30, Ref #5].
